# Targeted Therapies for Kirsten Rat Sarcoma (KRAS) G12C Mutant Metastatic Non-Small-Cell Lung Cancers

**DOI:** 10.3390/cancers15235582

**Published:** 2023-11-25

**Authors:** Cian O’Leary, Grace Murphy, Yong Yeung, Ming Tang, Vikram Jain, Connor G O’Leary

**Affiliations:** 1Mater Cancer Care Centre, Mater Hospital Brisbane, Brisbane, QLD 4101, Australia; 2Faculty of Medicine, Frazer Institute, University of Queensland, Translational Research Institute Australia, Brisbane, QLD 4072, Australia; 3School of Biomedical Sciences, Centre for Genomics and Personalised Health, Queensland University of Technology, Translational Research Institute Australia, Brisbane, QLD 4059, Australia; 4Faculty of Medicine, University of Queensland, Brisbane, QLD 4072, Australia

**Keywords:** NSCLC, KRAS, targeted therapy, sotorasib, adagrasib

## Abstract

**Simple Summary:**

Non-small-cell lung cancers (NSCLCs) are commonly diagnosed malignancies with poor prognosis. Kirsten rat sarcoma virus (KRAS) mutations are often observed in NSCLC and have classically been proven difficult to target. New targeted therapies for KRAS mutations (sotorasib, adagrasib) have become available to patients with NSCLC as of 2020. In this review, we assess the current evidence for these medications in terms of efficacy and safety, as well as future directions for these therapies.

**Abstract:**

Non-small-cell lung cancer (NSCLC) is a prevalent and often fatal malignancy. Advancements in targeted therapies have improved outcomes for NSCLC patients in the last decade. Kirsten rat sarcoma virus (KRAS) is a commonly mutated oncogene in NSCLC, contributing to tumorigenesis and proliferation. Though classically difficult to target, recently developed KRAS G12C inhibitors (sotorasib and adagrasib) have now overcome this therapeutic hurdle. We discuss the evidence for these medications, their pitfalls and adverse effects, as well as future directions in this space. Though these medications demonstrate substantial response rates in a heavily pre-treated advanced NSCLC cohort, as phase-3 evidence does not yet demonstrate an overall survival benefit versus standard-of-care chemotherapy, docetaxel. Additionally, these medications appear to have a negative interaction in combination with immunotherapies, with substantially greater hepatotoxicity rates observed. Despite this, it is undeniable that these medications represent an important advancement in targeted and personalised oncological treatment. Current and future trials assessing these medications in combination and through sequencing strategies will likely yield further clinically meaningful outcomes to guide treatment in this patient cohort.

## 1. Introduction

Lung cancer is the second most diagnosed cancer worldwide and the leading cause of cancer death, with non-small-cell lung cancer (NSCLC) accounting for over 80% of total cases [1]. Prior to 2002, cytotoxic chemotherapy was the only treatment for advanced NSCLC [2]. Prognosis was dismal, with 5-year overall survival rates of 16.9% across all stages in 2000 [3]. According to recent data, the availability and use of targeted therapies and immune checkpoint inhibitors have improved NSCLC prognoses significantly to a 5-year survival rate of 28% across all stages in 2023 [1]. The incidence of NSCLC has also been dropping worldwide over the last decade, largely due to declining rates of smoking [1]. Despite this, NSCLC remains a deadly condition, with significant associated morbidity and mortality for its sufferers. A large proportion of lung cancers present with advanced or metastatic disease at diagnosis, precluding curative options of treatment [1]. For the 30% or so of patients with a surgically resectable disease at diagnosis [1], over half may experience a recurrence of their disease post-operatively [4].

Improved understanding of the cell signalling pathways implicated in lung cancer tumorigenesis has led to developments in technologies allowing for the detection of actionable genetic mutations and, subsequently, the development of drugs to block these driver mutations. Lung cancers can accumulate a large mutational burden, and several targetable pathways have been identified for lung adenocarcinoma. These include EGFR, ALK, ROS1, HER2, MET, RET, BRAF, NTRK, and NRG1 fusions [5]. Systemic drug therapies targeting these pathways have yielded impressive and clinically significant improvements in outcomes, and some have even now replaced chemotherapy as first-line treatment strategies. The epidermal growth factor receptor (EGFR) mutation pathway is perhaps the most well-known success story highlighting this, with its first inhibitor, gefitinib, seeing Food and Drug Administration (FDA) approval in 2003 [6]. Since then, rapid advancement has been made in targeted therapy for this pathway, leading to the approval of osimertinib, a third-generation EGFR inhibitor, in 2018 for first-line management of advanced NSCLC [7]. Not only has osimertinib shown impressive OS and PFS benefits in the metastatic setting, but also its use in the adjuvant setting has now been shown to yield an OS benefit [8].

One common pathway, however, that has previously proved challenging to target directly has been mutations involving Kirsten rat sarcoma virus (KRAS). Previously considered impossible to target effectively, recent advances have led to the development of small-molecule-targeted inhibitors of this oncogene with significant implications for the future of NSCLC treatment in this space [9]. In this review, we will summarise the evidence supporting the use of these medications, as well as their toxicities, current practices for KRAS G12C NSCLC patients, and the future of clinical treatments in this space. While other reviews have covered this topic previously [9,10], we additionally present updates from previously reported trials, a summary of real-world data for these medications and novel therapeutic approaches currently under investigation.

## 2. Kirsten Rat Sarcoma Virus (KRAS)

KRAS is a commonly encountered mutated oncogene in human cancers [11]. Through production of the K-Ras protein (a guanine triphosphatase), it regulates cell signalling via the RAS/mitogen activated protein kinase (MAPK) pathway by cyclically binding to guanosine triphosphate (GTP) [11,12]. This represents KRAS’s active state, which acts as a cellular “on switch”, affecting downstream pathways involved in cellular growth and differentiation, before converting the GTP to guanosine diphosphate (GDP) and becoming inactive once again [11,12]. Mutations of KRAS are expressed in multiple tumour types, most notably in NSCLC at a rate of approximately 25–30% [13]. In its mutated form, KRAS remains GTP-bound and, as such, active due to decreased GTP hydrolysis, with increased activity resulting in tumour growth. In lifelong non-smokers, over half (56%) have the KRAS G12V mutation, where KRAS is kept in an activated GTP-bound state [13]. In smokers or ex-smokers, KRAS mutations typically occur on codon 12, with a substitution of glycine for cysteine at this location (G12C) being the most frequent, found in 42% of patients [13].

Among the KRAS mutant cancers, there is evidence to support that the G12C mutation confers a worse prognostic outcome. A multicentre retrospective review from Japan published in 2021 compared survival in patients with KRAS G12C mutant metastatic colorectal cancers and those with other KRAS mutations. It showed a significantly worse progression-free survival (PFS) and overall survival (OS) in the G12C cohort compared to those of other mutations (median PFS 9.4 versus 10.8 months, *p* = 0.015; median OS 21.1 versus 27.3 months, *p* = 0.015) [14]. In lung cancers, the prognostic significance of KRAS mutations has been the source of much debate, with some studies suggesting a shorter OS in KRAS mutant lung cancers but others suggesting the contrary. The 2015 TAILOR study demonstrated a significantly worse median overall survival between KRAS mutant and KRAS wildtype patients with advanced NSCLC who had previously been treated with platinum-based chemotherapy [15]. As such, the presence of a KRAS mutation confers an overall negative prognostic outcome for NSCLC patients and represented an area of unmet need in targeted therapy for many decades.

It is becoming increasingly apparent that complex KRAS downstream interactions and co-mutations also influence tumour signalling in lung cancer. There are three co-mutations of particular interest in the recent literature: TP53, Kelch-like ECH-associating protein 1 (KEAP1), and STK11/NRF2, of which the latter two are considered equivalent mutations. KRASG12C has also been associated with ERBB2 and ERBB4 mutations [16]. Co-mutation with TP53 occurs in 38–42% of patients, with KEAP1 in 8.1–27%, and with STK11 11.8–29% of patients [16,17,18,19], with some variability as to the frequency of KEAP1 and STK11. TP53 co-mutations were independently associated with high PD-L1 expression (odds ratio (OR), 6.36; 95% confidence interval (CI), 1.84–22.02; *p* = 0.004) [18]. STK11/NRF2 protects cells against carcinogens and oxidants. STK11/NRF2 is a transcription factor that mediates the induction of phase-2 detoxification enzymes. Kelch-like ECH-associating protein 1 (KEAP1) binds to NRF2 and represses NRF2 transcriptional activity [20]. In one study, KEAP1 and STK11 had no association with PD-L1 status, while another suggested it may be associated with a low PD-L1 expression but a high TMB score [18].

KRAS co-mutations have been studied to determine their effects on prognosis in lung cancer patients. It is consistent that TP53, while repeatedly associated with a poorer prognosis in other tumour streams, has no association with decreased overall survival in KRAS mutant lung cancer [17,18]. STK11 and KEAP1 have been associated with a poorer prognosis in KRAS mutant lung cancer [17,18]. Both STK11 and KEAP1 have been associated with shorter overall survival under platinum-based chemotherapy [18]. In Arbour et al.’s study using multivariate analysis only, KEAP1 appeared to effect prognosis in KRAS mutant lung cancer (hazard ratio (HR), 1.96; 95% CI, 1.33–2.92; *p* < 0.001), and the authors argued that the high proportion in concurrent KEAP1 and STK11 mutations may have skewed the univariate analysis results for STK11. In this same review, KEAP1 was associated with a decreased overall survival under both chemotherapy and immunotherapy, while there was no effect from TP53 or STK11 mutations [17]. STK11 and KEAP1 mutations do not seem to influence the prognosis in KRAS wildtype lung cancers [19].

## 3. KRAS Detection

As new treatment options for patients with KRAS-mutated lung cancers become available, the need for improved methods of testing for KRAS mutations becomes more important [21].

Previously, KRAS testing was performed on tumour tissue, where DNA is extracted from formalin-fixed paraffin-embedded (FFPE) tissue blocks, followed by a variety of polymerase chain reaction (PCR)-based testing methods including sanger sequencing and pyrosequencing. Sanger sequencing of PCR-amplified DNA was previously considered the gold-standard technique. One criticism of this method is its relatively modest limit of detection. Multiple studies have demonstrated that the Sanger sequencing method requires a minimum of 15% to 50% of the sample DNA to contain the KRAS mutation before reliable detection is achieved [22]. As sequencing technologies can lack sufficient sensitivity, efforts have been made to look into alternative methods of KRAS testing.

Whilst Sanger sequencing and next-generation sequencing (NGS) share some key principles, NGS provides a much higher sequencing volume through its ability to process millions of reactions in parallel. This results in a high-throughput, high-sensitivity method of testing, allowing for testing with more rapid turnarounds at a reduced cost, with many genome sequencing projects which would have taken many years with Sanger sequencing completed within hours using NGS [23]. Despite this, NGS-based assays for genomic alterations in tumours can still take up to 12–15 days [24], a suboptimal timeframe for the treatment of NSCLC. Moreover, due to the complexity of sample processing for NGS, bottlenecks can occur in the management, analysis, and storage of datasets [23].

Recently, the Idylla oncology assay by Biocartis was launched to complement NGS testing. Currently, the Idylla platform enables the detection of KRAS hotspot mutations, along with BRAF, EGFR, and NRAS mutations with rapid turnaround times of less than 3 h. Unlike NGS-based assays, the Idylla system uses formalin-fixed paraffin-embedded (FFPE) tissue samples without the need for DNA extraction with a fully automated interpretation of the results [24]. Moreover, several studies have demonstrated and confirmed the improved validity, accuracy, and concordance of the Idylla system in comparison to those of NGS in detecting KRAS, EGFR, BRAF, and NRAS hotspot mutations.

## 4. Current Evidence

The development of targeted therapies against KRAS has proven challenging for a number of reasons, namely, its strong binding affinity for GTP, which is widely abundant in cytoplasm [25]. Additionally, KRAS in a particularly smooth and small molecule, with few potential binding sites, further complicating advancements in targeted therapy [10]. This differs from other commonly expressed lung cancer mutations such as in ROS1 [26], anaplastic lymphoma kinase (ALK) [27], and the epidermal growth factor receptor (EGFR) [28] which have all seen significant success and acceleration with targeted therapy development in the last decade. Recent breakthroughs, however, have led to the development of targeted inhibitors of KRAS G12C: sotorasib and adagrasib. These molecules can selectively bind to the cysteine 12 protein in the switch-II pocket of KRAS when it is GDP-bound and keep it in its inactive form [29,30], as described in Figure 1. The current evidence for these is described below and summarized in Table 1 and Table 2.

## 5. Sotorasib

Sotorasib was the first inhibitor of KRAS G12C approved by the FDA [35] and, subsequently, the Therapeutic Goods Administration (TGA) in Australia [36] for use in NSCLC following trials published in 2020 and 2021. In CodeBreak 100, a phase I-II trial, patients with KRAS G12C mutant cancers (mainly NSCLC, but also other malignancies, most notably colorectal) received oral sotorasib, with an aim to assess its safety and efficacy [31]. In total, 129 patients with pre-treated (median of three prior lines of treatment) advanced cancers were enrolled to receive either 180, 360, 720, or 960 mg of sotorasib as part of dose escalation/expansion cohorts in phase I. Of the 59 patients included with NSCLC, the majority had previously received anti-programmed cell death ligand-1 (PDL-1) therapy (89.8%), and all had received prior platinum-based therapy. Over half of the patients in the trial had a treatment-related adverse event of any grade (n = 73, 56.6%), though notably, no dose-limiting adverse events or deaths related to adverse events were reported. Gastrointestinal upset and fatigue constituted the majority of the reported adverse events, with grade 3–4 events (n = 15, 11.6%) largely taking the form of derangement of one or more liver function enzymes. For the NSCLC group, almost one third of patients had a confirmed response to treatment (32.2%), most notably in the cohort receiving 960 mg daily of sotorasib (though responses were seen across all dose levels), with a median duration of response of 4 months and median progression-free survival (PFS) of 6 months [31].

On the back of this phase-I data, phase II of CodeBreak 100 was implemented to further assess the efficacy of sotorasib in NSCLC [32]. In this single-group phase-2 trial, patients pre-treated (previous platinum-based therapy and/or anti-PDL1 therapy) for KRAS G12C mutant lung cancer received second-/third-line sotorasib, with a primary endpoint of objective response (OR). Overall, 126 patients were enrolled and received 960 mg of oral sotorasib once daily. Over two thirds of these patients received sotorasib for at least 3 months, with almost one third continuing out to 9 months. The OR rate (ORR) was 37.1%, with a median duration of response of 11.1 months. Interestingly, response rates were observed across all subgroups of PDL-1 expression. Similarly, substantial PFS and overall survival (OS) (6 and 12.5 months, respectively) were also noted. Approximately 70% of patients had an adverse event attributable to sotorasib, with gastrointestinal upset, liver function test derangement, and fatigue being the most common adverse events across all grades. Dose reduction/interruption was required in 22.2% of all patients due to treatment-related adverse events [32]. Despite these issues, due to the clinical efficacy demonstrated, this study was subsequently incorporated into the National Cancer Care Network (NCCN) guidelines, supporting the use of sotorasib in patients with NSCLC as a second-line agent [37]. Sotorasib received accelerated FDA approval in 2021 [35].

Sotorasib’s PFS benefit was confirmed in the CodeBreak 200 study, a randomized open-label phase-3 trial by de Langen et al. from February 2023 comparing sotorasib versus docetaxel in a similarly pre-treated NSCLC population [33]. Patients were randomized to receive sotorasib 960 mg OD (n = 171) versus a triweekly infusion of 75 mg/m^2^ of docetaxel (n = 174). Patients were excluded if they had concurrent EGFR mutations or ALK translocations. The primary endpoint was PFS, and patients continued therapy until they were confirmed to have progressive disease, death, or intolerance of the treatment itself. Patients receiving sotorasib had longer median PFS rates (5.6 months (95% CI 4.3–7.8) versus 4.5 months (95% CI 3.0–5.7), HR 0.66 [0.51–0.86], *p* = 0.0017) compared to those of docetaxel patients and were 34% less likely to progress in their disease. Patients also had double the overall response rate with sotorasib (28.1% vs. 13.2%). Patients with prior intracranial metastases were also found to have a longer time until progression, 15.8 months for sotorasib vs. 10.5 months for docetaxel (HR, 0.52; 95% CI, 0·26–1·0). In the intended treatment population, there was no difference in overall survival (HR, 1.01 [0.77–1.33]).). The lack of an overall survival benefit might have been influenced by the 34% of patients in the docetaxel arm receiving a subsequent KRASG12C inhibitor as well as the 13% vs. 1% drop-out rate in the docetaxel arm. The authors note that the patients who dropped out of the docetaxel arm had poorer demographic features which, when not included, could have narrowed any difference in overall survival. There was a more favourable safety profile (n = 56 grade 3 or greater events versus n = 61) with sotorasib, and the phase-3 toxicity profile was similar to that in the phase-2 study. A total of 15% of patients required a dose reduction, 10% discontinued sotorasib treatment, and one patient died following treatment-related adverse events. Notably, higher rates of grade-3 adverse events and liver function derangement were noted in patients who received immunotherapy within 2.6 months of starting sotorasib. While there was no overall survival benefit with sotorasib compared to docetaxel, its toxicity profile and intracranial activity present it as a favourable option over docetaxel in this cohort of patients [33].

## 6. Adagrasib

In addition to sotorasib, a second KRAS G12C inhibitor, adagrasib, has also demonstrated efficacy in the recently published 2022 KRYSTAL-1 phase-2 trial. Following on from a dose-escalation assessment in their phase-1 trial [38], patients with unresectable or metastatic NSCLC with a KRAS G12C mutation were treated twice daily with a 600 mg dose of oral adagrasib, with a primary endpoint of objective response [34]. These patients, similar to in the above sotorasib trials, had to have been previously treated with both platinum-based chemotherapy and an immune checkpoint inhibitor. Overall, 116 patients were enrolled and 112 had measurable disease at baseline. Among the latter, the objective response rate was 42.9% (95% CI, 33.5–52.6), with one having a complete response, forty-seven having a partial response, and forty-one having stable disease. The median duration of disease response was 8.5 months, with a median PFS of 6.5 months and a median OS of 12.6 months. Diarrhoea was the most common adverse event seen, occurring in 70.7% of all patients, followed by nausea, fatigue, and vomiting. Events of grade 3 or higher occurred in 44.8% (n = 52) of patients, with fatigue, nausea, and liver function derangement (raised ALT and AST) being the most common. Just over half of all patients required a dose reduction due to an adverse event (51.7%, n = 60), again most commonly being attributed to gastrointestinal upset or liver function derangement. Two patients died during the trial: one of cardiac failure in the setting of a known pericardial effusion and the other with a pulmonary haemorrhage. It was noted that while over 90% of patients had GIT toxicity within the first cycles, this, on average, resolved in 2 weeks and significantly declined from cycle 3 onwards. The higher toxicity profile may be due to the much longer half-life of 23 h with adagrasib compared to 5 h with sotorasib. KRYSTAL-1 also reported the intracranial response. A total of 33.3% patients had an intracranial response with an intracranial PFS of 5.4 months in 42 included patients [34]. On the basis of these findings, adagrasib was approved by the FDA in the United States in 2022 [39]. The recently published 2-year follow-up data from KRYSTAL-1 has demonstrated a durable response, with the median OS now extending out to 14.1 months and a median 2-year survival rate of 31% [40]. Similarly, another recent update from this trial has demonstrated the efficacy of adagrasib in patients with untreated intracranial disease (n = 25), with comparable a ORR and PFS to those of the main cohort (42% and 5.4 months, respectively). This cohort was excluded in the CodeBreak trials with sotorasib, making this a novel finding for this class of drugs [41]. The phase-3 KRYSTAL-12 trial of adagrasib versus docetaxel will further investigate this effect [42].

## 7. Real World Evidence and Outcomes

Real world data complement those collected in a clinical trial and provide practical and relatable insights into how drugs work, outside the more stringent confines of a randomized controlled trial setting. They add to the body of evidence created by a trial by providing insight into outcomes for more heterogenous populations, under more practical application. The data demonstrated in the CodeBreak trials have been corroborated by real world data from international centres (summarized in Table 3 below). A large multicentre report from France using data gleaned from the sotorasib early-access program had comparable ORRs and PFSs to those in the CodeBreak data (34.4% and 4.2 months, respectively) [43]. This was a similarly heavily pre-treated population, with a median of two prior lines of therapy. In the UK, a 2022 report of 89 patients across 22 centres receiving second-line sotorasib also demonstrated similar ORRs and PFSs to those of CodeBreak (34.8% and 185 days, respectively) [44]. Fewer high-grade adverse events were noted than in the CodeBreak data (9% vs. 20%) with diarrhoea being the most common adverse event reported, though the rate of dose reductions due to adverse events was similar (19.1% vs. 22.2%) [44]. A similar retrospective review from the US (specifically, centres in New York) of 105 patients with NSCLC receiving sotorasib showed a median PFS of 5.3 months (or approximately 159 days) and an ORR of 28% [45]. Data of 173 German patients treated with sotorasib reported an ORR of 38.7% with an OS of 9.8 months and an intracranial real-world PFS of 7.5 months in the 38% of patients with brain metastases in the cohort [46]. A report from a French institution showed an ORR of 47% and a median PFS of 5.5 months, though the sample size was, notably, much smaller (n = 15) [47].

The temporal relationship between immunotherapy and higher-grade adverse events on G12C inhibitors has also been observed in real-world data. The above report by Thummalapalli et al. reported grade-3 adverse-event rates of up to 28% in patients with recent anti-PDL1 therapy receiving sotorasib, up to 3 months from the last dose of immunotherapy [45]. No grade 3+ events were noted in patients who were greater than 12 weeks from their last immunotherapy dose, and additionally, the rate was much lower in those who were PDL-1-naïve (5%, n = 1/19). A history of immune-related adverse events prior to sotorasib initiation was not associated with a greater risk of high-grade toxicity on sotorasib [45]. Further US data have demonstrated this association, with higher rates of hepatotoxicity in patients who had received immunotherapy within 90 days prior to their sotorasib start date [48]. One French report demonstrated significantly greater rates of all grade-3-or-greater adverse events in patients sequentially prescribed sotorasib within 30 days of their last immunotherapy [49]. In this review, 102 patients with KRAS G12C mutant NSCLC who received sotorasib were retrospectively enrolled. Almost half (47%, n = 48) received immunotherapy as their most recent line of therapy prior to enrolment, with the median time from their last immunotherapy infusion being 1.6 months for this group. Of the remainder, the majority had received immunotherapy at some point in their prior care (83%, n = 45). The cohort of patients that sequentially changed from immunotherapies to sotorasib had substantially greater rates of grade 3+ events (50% versus 13%, *p* < 0.001), with liver function derangement again being the most commonly reported in the recent immunotherapy cohort (33% versus 11%, *p* = 0.006). These grade-3-or-greater events were more common within 30 days and, to a lesser degree, up to 60 days from their last immunotherapy treatment. Dose modifications and discontinuation occurred in about one third of this cohort (38% and 31%, respectively) [49].

This window for increased toxicity is congruent with the CodeBreak trial data and presents an important clinical dilemma for clinicians going forward. Delaying the commencement of second-line therapy for 3 months after a recent progression of disease is not feasible in clinical practice; however, 30 days, as suggested by Chour et al., may be a more palatable timeframe in select clinical situations [49]. Additionally, there are no data to date to our knowledge that any other patient factors aside from this temporal association increase this risk of hepatic injury. Alcohol intake and prior liver metastases were not noted to impact this risk significantly by Chour et al. [49]. Prior immune-related adverse events did not contribute to toxicity from sotorasib in Thummalapalli et al.’s report [45]. Additionally, real-world reports suggest that this hepatotoxicity from sotorasib is readily manageable with steroids [45,49]. This is coupled with the improved reported patient quality of life outcomes on sotorasib noted in CodeBreak 200 [33]. Accounting for its known PFS benefit over docetaxel, it is difficult to justify avoiding sotorasib even with this association, though a 30-day delay in commencement may be considered. However, as there were no OS differences in the second line as a monotherapy as seen in CodeBreak 200, docetaxel remains a reasonable option in those patients who recently progressed on immunotherapy.

## 8. Mechanisms of Resistance

Unlike prior T790 mutations seen with earlier EGFR tyrosine kinase inhibitors (TKI), the mechanisms of resistance to KRAS G12C inhibitors are still being elucidated and are likely to be multi-modal. Mutations at the switch-II binding pocket where both sotorasib and adagrasib bind have been seen [50]. Mutations at Y96C, R68S, and H95D/Q/R have been observed post-exposure to adagrasib [50]. In an X-ray crystallographic analysis of the conformational binding of adagrasib and sotorasib, only adagrasib was affected by H95D/Q/R mutations, suggesting a possible role for drug switching. However, it was also noted that patients who had this H95D/Q/R mutation also had co-mutations that might also render sotorasib ineffective. Non-binding sites that may also confer resistance are mutations at codons 13, 59, 61, 117, and 146 which, when involved, would impede GTP hydrolysis or facilitate GDP-to-GTP nucleotide exchange [50]. KRAS G12C resistance has also been associated with a mutation elsewhere in the RTK-RAS pathway. These include mutations of NRAS, BRAF, and MAPK [50,51]. Given this, current and future studies assessing KRAS inhibition in combination with other medications known to affect the RAS-RAF pathway (SHP2 inhibitors, tyrosine kinase inhibitors) may prove fruitful in overcoming these resistances. In this context, improvements in DNA sequencing may play an even more substantial role in the tailored treatment of NSCLC patients, detecting targetable adaptive mutations in biopsy samples post-progression on a KRAS G12C inhibitor.

Evidence from clinical trials also suggests a resistance mechanism in patients with KEAP1 co-mutations. In data from KRYSTAL-1 and phase 2 of CodeBreak 100, patients with KEAP1 mutations demonstrated poorer overall clinical outcomes comparatively [32,34]. Real-world data from the USA and Germany also suggest that KEAP1 co-mutations in patients receiving sotorasib are associated with resistance [45,46]. A large retrospective review from Germany demonstrated greater rates of disease progression at one year in patients with a KEAP1 co-mutation (HR 2.0, 0.9–4.2, *p* = 0.01) [46]. Similarly, a retrospective review from the USA demonstrated a shorter PFS (2.0 versus 5.3 months, HR 3.19, *p* = 0.04) and OS (5.2 vs. 12.6 months, HR 4.1, *p* = 0.003) comparatively in patients with this mutation [45].

## 9. Future Directions

In the future of these compounds in clinical use, we will see them explored as monotherapies (in advanced and early clinical settings), as combinations with other anti-cancer therapies, and as newer and more potent drugs. Currently there are a multitude of clinical trials involving KRAS G12C inhibitors across these parameters collecting data, and these are summarized in Table 4 and Table 5 below. As a follow-up on KRYSTAL-1, adagrasib has now entered phase-3 trials in the form of the KRYSTAL-12 study. Similar to CodeBreak 200, this will involve patients with pre-treated KRAS G12C mutant metastatic NSCLC randomized to receive adagrasib versus docetaxel [42]. Newer generations of KRAS G12C inhibitors are also undergoing assessments for efficacy and safety. A phase-1 trial, KontRASt-01, assessing a new KRAS G12C inhibitor molecule JDQ443, aims to look at the clinical efficacy of this compound both alone and in combination with tislelizumab and/or TNO155 (an src homology-2 domain inhibitor) and has been recruiting since early 2021 [52]. Recent safety and efficacy updates of JDQ443 monotherapy from this trial have been promising, with an ORR thus far of 41.7% across all dose levels in the NSCLC cohort (n = 10 at the time of the update). Its toxicities appear similar to those of sotorasib and adagrasib with fatigue and gastrointestinal upset being the most commonly reported, though this new molecule additionally appears to cause a peripheral neuropathy [52]. Garsorasib (D-1553), another newer KRAS G12C inhibitor, has also shown a promising ORR of 40.5% in a population of 79 patients [53].

There are a multitude of phase-I dose-escalation studies (e.g., CodeBreak 101) to find the best KRAS G12C targeting molecule as well as to develop a pan-KRAS targeted therapy [54]. These studies are reflective of the current body of evidence for KRAS G12C inhibitors and their use in the metastatic setting, seeking to either confirm a PFS/OS benefit versus the standard of care or ascertain new and potentially efficacious combinations with these drugs. Early data from the carboplatin and pemetrexed combination arm of CodeBreak 101 have yielded interesting clinical information not only for second-line therapy but for first-line therapy, as well. This so far is a small cohort of patients (n = 30) receiving sotorasib 960 mg daily, pemetrexed 500 mg/m^2^ and up to four cycles of carboplatin. Most patients were treatment-naïve (63%, n = 19). The ORR (confirmed and unconfirmed) was higher in the first-line setting, at 73% compared to 55% in those who had received prior treatment lines. The disease control rate in the first-line setting was also 100%. Reported toxicity rates were also quite elevated with a grade-3–4 adverse-event rate of 63% across all patients, with myelosuppression being the most commonly reported event [55].

**Table 4 cancers-15-05582-t004:** A summary of clinical trials currently underway exploring KRAS G12C inhibitors versus chemotherapy in NSCLC.

Phase III Trials	Intervention	Control	Sponsor
Phase 3 Study of MRTX849 (Adagrasib) vs. Docetaxel in Patients With Advanced Non-Small Cell Lung Cancer With KRAS G12C Mutation (KRYSTAL-12) [42]	Adagrasib	Docetaxel	Mirati Therapeutics (San Diego, CA, USA)
Study of JDQ443 in Comparison with Docetaxel in Participants With Locally Advanced or Metastatic KRAS G12C Mutant Non-small Cell Lung Cancer (KontRASt-02) [56]	JQD443	Docetaxel	Novartis Pharmaceuticals (Basel, Switzerland)
Study to Compare AMG 510 “Proposed INN Sotorasib” With Docetaxel in Non Small Cell Lung Cancer (NSCLC) (CodeBreak 200) [57]	Sotorasib	Docetaxel	Amgen (Thousand Oaks, CA, USA)

**Table 5 cancers-15-05582-t005:** A summary of current trials assessing KRAS G12C inhibitors in combination with immunotherapies and/or chemotherapies.

Trial	Phase	Treatment	Sponsor
Efficacy and Safety of IBI351 in Combination with Sintilimab ± Chemotherapy in Advanced Non-squamous Non-small Cell Lung Cancer Subjects with KRAS G12C Mutation [58]	I	Sintilimab +/− chemotherapy +/− IBI351	Innovent Biologics (Suzhou, China) Co., Ltd.
A Study to Evaluate D-1553 in Combination Therapy in Non-Small Cell Lung Cancer [59]	I/II	D-1553 with immunotherapy or targeted	InventisBio Co., Ltd. (Shanghai, China)
KRAS-Targeted Vaccine with Nivolumab and Ipilimumab for Patients with NSCLC [60]	I	KRAS peptide vaccine with poly-ICLC adjuvant with Ipilimumab (1 mg/kg), nivolumab (3 mg/kg)	Sidney Kimmel Comprehensive Cancer Center at Johns Hopkins (Baltimore, MD, USA)
A Study to Evaluate the Safety, Pharmacokinetics, and Activity of GDC-6036 Alone or in Combination in Participants with Advanced or Metastatic Solid Tumors with a KRAS G12C Mutation [61]	I/II	GDC 6036 + pembrolizumab	Genentech, Inc. (San Francisco, CA, USA)
Study of JDQ443 in Patients with Advanced Solid Tumors Harboring the KRAS G12C Mutation (KontRASt-01) [62]	I/II	JDQ443 in combination with tislelizumab +/− TNO155 (SHP2 inhibitor)	Novartis Pharmaceuticals (Basel, Switzerland)
A Study of MK-1084 as Monotherapy and in Combination with Pembrolizumab (MK-3475) in Participants with KRAS G12C Mutant Advanced Solid Tumors (MK-1084-001) [63]	I	MK-1484 +/− pembrolizumab	Merck Sharp & Dohme LLC (Rahway, NJ, USA)
A Study Evaluating the Safety, Activity, and Pharmacokinetics of GDC-6036 in Combination with Other Anti-Cancer Therapies in Participants with Previously Untreated Advanced or Metastatic Non-Small Cell Lung Cancer with a KRAS G12C Mutation [64]	I/II	GDC 6036 + pembrolizumab	Hoffmann-La Roche (Basel, Switzerland)
Sotorasib Activity in Subjects with Advanced Solid Tumors with KRAS p.G12C Mutation (CodeBreak 101) [65]	I/II	Sotorasib in various combinations including pembrolizumab, atezolizumab or pembrolizumab with chemotherapy	Amgen (Thousand Oaks, CA, USA)
Combination of CAR-DC Vaccine and ICIs in Local Advanced/Metastatic Solid Tumors [66]	I	CAR-DC vaccine, abraxane, cyclophosphamide, Ipilimumab, PD-L1 immunotherapy	Chinese PLA General Hospital (Beijing, China)
Combination Therapies with Adagrasib in Patients With Advanced NSCLC With KRAS G12C Mutation [67]	II	Cohort A: PD-L1 TPS > 1% Adagrasib + Pembrolizumab Cohort B: Adagrasib + Pemetrexed + Pembrolizumab Cohort C: Adagrasib + Platinum + Pemetrexed + Pembrolizumab 4 cycles, then maintenance Adagrasib + Pemetrexed + Pembrolizumab	Mirati Therapeutics Inc. (San Diego, CA, USA)
A Phase I Study of Adagrasib and Durvalumab for Treatment of Advanced Non-small Cell Lung Cancers and Gastro-intestinal Cancers Harboring KRAS G12C Mutations [68]	I	Adagrasib + Durvalumab	M.D. Anderson Cancer Center (Houston, TX, USA)
Safety and Efficacy Study of SAR442720 in Combination with Other Agents in Advanced Malignancies [69]	III	SAR442720 + Pembrolizumab	Sanofi (Paris, French)
A Study Evaluating Sotorasib Platinum Doublet Combination Versus Pembrolizumab Platinum Doublet Combination as a Front-Line Therapy in Participants with Stage IV or Advanced Stage IIIB/C Nonsquamous Non-Small Cell Lung Cancers (CodeBreaK 202) [70]	III	Carboplatin & pemtrexed in combination with either Sotorasib or Pembrolizumab	Amgen (Thousand Oaks, CA, USA)
Phase 2 Trial of MRTX849 Monotherapy and in Combination with Pembrolizumab and a Phase 3 Trial of Adagrasib in Combination in Patients With a KRAS G12C Mutation KRYSTAL-7 [71]	II/III	Adagrasib +/− Pembrolizumab In both PDL1 TPS <1% or >1% cohorts	Mirati Therapeutics Inc. (San Diego, CA, USA)

An aforementioned point of concern is the increased rate of higher-grade adverse events in patients previously treated with immunotherapy [33]. Immunotherapies have changed the landscape of NSCLC management, significantly improving OS rates in both early and advanced disease [72,73,74]. Early evidence, again from CodeBreak 101, has shown increased rates of grade-3-or-higher adverse events in patients receiving combination sotorasib and anti-PDL1 medication with 60–80% of the patients in combination groups experiencing a grade-3-or-higher event [75]. By utilizing a lead-in period with sotorasib prior to the commencement of immunotherapy, this adverse-event rate was reduced (by 50% in patients receiving atezolizumab and 33% in those receiving pembrolizumab) [75]. Similar to the CodeBreak 200 data results, the major adverse events demonstrated have been liver function derangement and gastrointestinal upset [33,75]. It is notable that despite these adverse events, there have been durable response rates of 17.9 months and overall survival rates of 15.7 months, suggesting a positive synergy in terms of tumour response. How these two aspects will balance going forward remains unclear, though both clinical trial data and real-world data have highlighted this toxicity as a point of concern for future research and clinical practice [75].

An interesting space for targeted therapies like these is in the neoadjuvant or adjuvant setting. The benefit of early targeted therapy for patients with actionable mutations has been seen, as previously mentioned, in the 2020 ADAURA trial of osimertinib [8]. Early phase-2 trials are already underway in the United States assessing the efficacy of both sotorasib and adagrasib in the neoadjuvant setting for KRAS G12C mutant lung cancers, including neoadjuvant therapy for 6 weeks of adagrasib with placebo or nivolumab (NCT05472623) [76] and neoadjuvant therapy for four cycles of platinum-based chemotherapy and sotorasib (NCT05118854) [77]. It is worth noting that the patient cohorts included in the CodeBreak and KRYSTAL trials were heavily pre-treated [33,34]. They represent a cohort of patients with more advanced disease, potentially with cancer that is not particularly treatment responsive. It is possible that sotorasib and adagrasib may have more efficacy in treatment-naïve cases, both in the curative and advanced setting.

Aside from direct inhibitors of KRAS G12C, other approaches aimed at treating these mutant lung cancers via downstream regulation are under investigation. Signalling pathways that modulate KRAS activity include RAF-MEK-ERK/MAPK and PI3K-AKT-mTOR, and additional molecules that interplay with these include SHP2 and CDK 4/6 [9]. In particular, the MEK/ERK/MAPK pathway seems to be a popular target in the current space for these trials. Trametinib, a MEK/MAPK inhibitor, is currently being investigated in combination with pembrolizumab in patients with KRAS mutant NSCLC [78]. Anlotinib, a tyrosine kinase inhibitor, has been shown to exert an anti-cancer effect in vitro on KRAS mutant lung cancer cells by inhibiting the MEK/ERK pathway [79]. A phase-1 trial is now recruiting, assessing anlotinib in combination with trametinib in patients with advanced KRAS mutant NSCLC. A more novel approach is being explored with DCC-3116, an inhibitor of unc-51-like autophagy activating kinase (ULK). ULK 1 and 2 are implicated in cancer cell survival by initiating autophagy and are additionally modulated by the MAPK pathway. This phase-1 trial looks to use DCC-3116 alone and in combination with trametinib in RAS- and RAF-mutated malignancies [80]. Inhibitors of these downstream signalling pathways may potentially also help overcome resistance mechanisms to KRAS G12C inhibitors as combinations. SHP2 inhibition has been demonstrated to remove adaptive resistances to KRAS G12C inhibition in vitro [81]. KontRASt-01 is examining G12C and SHP2 inhibition in combination, with early data from the NSCLC cohort (n = 12) demonstrating both clinical responses (33.3%) and disease control (66.7%) to date. These 12 patients had already received treatment with KRAS G12C inhibitors, suggesting an overcoming of drug resistance and providing a clinical correlate for prior in vitro data [82].

## 10. Conclusions

Despite advances and improvements in preventative strategies and treatment options, NSCLC remains a challenging malignancy to effectively treat, with significant morbidity and mortality associated for affected patients. KRAS G12C inhibitors represent an exciting new advancement in NSCLC treatment as a potential alternative to docetaxel in patients with advanced/pre-treated disease. Following on from other more classic targeted therapies, such as EGFR inhibitors, it is likely that these medications will see increasing trial use in both the palliative and curative settings, though their ultimate impact on the state of NSCLC treatment is still uncertain. Fleshing out synergies between these medications and current standard-of-care systemic treatments may further improve outcomes. As our ability to detect and target actionable mutations continues to improve, so too will outcomes for patients within these at-risk cohorts.

## Figures and Tables

**Figure 1 cancers-15-05582-f001:**
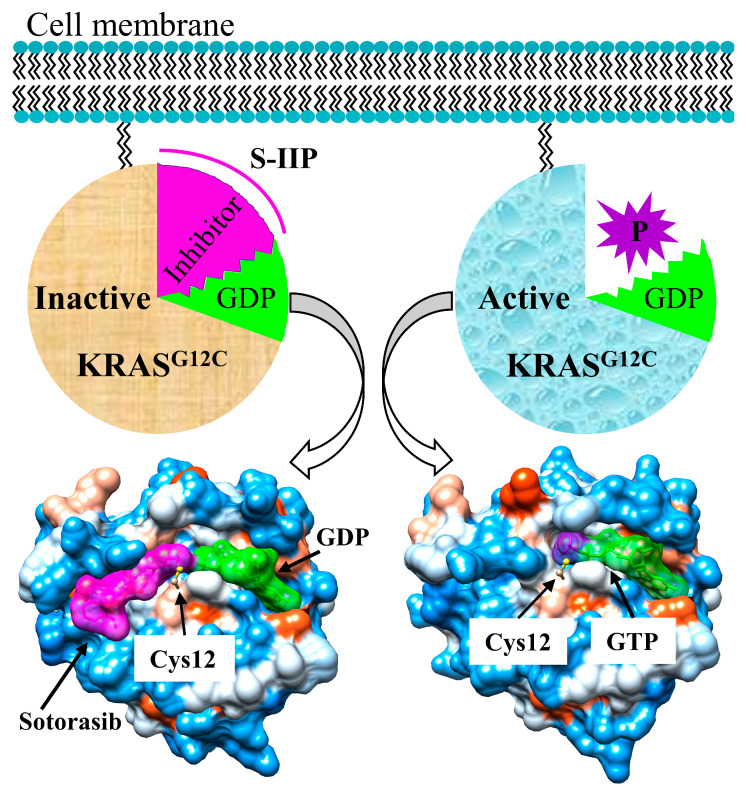
Schematic diagram illustrating the mechanism that defines how KRAS^G12C^ inhibitors (in magenta) lock KRAS^G12C^ in its inactive states. KRAS^G12C^ inhibitors such as sotorasib and adagrasib covalently bind to residue Cys12 (in ball and stick) in the switch-II pocket of KRAS^G12C^, which inhibits the binding of phosphate (in purple) to the switch-II pocket of KRAS^G12C^ and prevents the production of GPT from GDP (in green).

**Table 1 cancers-15-05582-t001:** Reported efficacy outcomes from phase 1–3 trials for sotorasib and adagrasib. n = number of patients in study population, ORR = objective response rate, DoR = duration of response, OS = overall survival, PFS = progression free survival.

Trial	n	Drug	Efficacy
CodeBreak 100 (Phase 1) [31]	129	Sotorasib	ORR 32%
CodeBreak 100 (Phase 2) [32]	126	Sotorasib	ORR 37.1%
			DoR 11.1 months
			OS 12.5 months
CodeBreak 200 (Phase 3) [33]	345	1:1 Sotorasib vs. docetaxel	PFS 5.6 months vs. 4.5, *p* = 0.0017
			ORR 28.1% vs. 13.2%
			No OS benefit
KRYSTAL-1 (Phase 2) [34]	116	Adagrasib	ORR 42.9%
			DoR 8.5 months
			PFS 6.5 months
			OS 12.6 months

**Table 2 cancers-15-05582-t002:** Toxicity of KRAS G12C inhibitors from phase 1–3 trials. n = number of patients in study population, ALT = alanine aminotransferase, AST = aspartate aminotransferase.

Trial	n	Drug	Grade 3+ Adverse Events for G12C Inhibitor
CodeBreak 100 (Phase 1) [31]	129	Sotorasib	ALT rise 4.7%
			Anaemia 4.7%
			Vomiting/ Diarrhoea 3.9%
CodeBreak 100 (Phase 2) [32]	126	Sotorasib	AST rise 6.3%
			ALT rise 5.6%
			Diarrhoea 4%
CodeBreak 200 (Phase 3) [33]	345	1:1 Sotorasib vs. docetaxel	Diarrhoea 12%
			ALT rise 8%
			AST rise 5%
KRYSTAL-1 (Phase 2) [34]	116	Adagrasib	Anaemia 5.2%
			Nausea 4.3%
			Fatigue 4.3%
			AST/ALT rise 3.4%

**Table 3 cancers-15-05582-t003:** A summary of real-world data of sotorasib efficacy and safety. n = number of patients. IO = immunotherapy. * = only hepatotoxicity events reported.

Author	n	Prior IO	Efficacy	Grade 3 Adverse Event Rate
Bessy et al. (France) [47]	15	Not reported	ORR 47% PFS 5.5 months	20%
Althoff et al. (Germany) [46]	173	Not reported	ORR 38.7% OS 9.8 months	Not reported
Wislez et al. (France) [43]	313	164/313	ORR 34.35 PFS 4.2 months	7% *
Julve et al. (UK) [44]	89	Not reported	ORR 34.8% PFS 185 days	9%
Thumallapalli et al. (USA) [45]	105	86/105	ORR 28% PFS 5.3 months	16%
Desai et al. (USA) [48]	31	28/31	ORR not reported PFS 3.3 months	31%
Chour et al. (France) [49]	102	93/102	Not reported	31%

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
