# Peer review of "Targeted Therapies for Kirsten Rat Sarcoma (KRAS) G12C Mutant Metastatic Non-Small-Cell Lung Cancers"

_cancers, 2023, doi:10.3390/cancers15235582_

Round 1
Reviewer 1 Report
Comments and Suggestions for Authors
None
Author Response
Many thanks for your feedback. We have made significant updates to the paper as outlined in our cover letter which we feel address the main comments from reviewers. We thank you for your ongoing input.
Reviewer 2 Report
Comments and Suggestions for Authors
In this manuscript, the authors provide an overview of crucial role of KRAS mutation in the pathogenesis and progression of NSCLC and focus on the current state of knowledge on the progress made in the clinical development of two KRAS inhibitors, sotorasib and adagrasib. Most of the information included in this manuscript was presented in the articles by Huang, L. et al. (Signal Transduct Target Ther. 2021 Nov 15;6(1):386. doi: 10.1038/s41392-021-00780-4) and Kwan, A.K. (J Exp Clin Cancer Res. 2022 Jan 19;41(1):27. doi: 10.1186/s13046-021-02225-w.). Moreover, this manuscript is limited to the small molecule synthetic KRAS inhibitors without discussing of other target therapeutic strategy that suppresses KRAS activity through downregulation of KRAS signaling pathways. Overall, this review manuscript does not provide any new information on and valuable insight into the development of KRAS targeted therapy, thereby making limited contribution to the field.
Author Response
Many thanks for the comments. Please note we agreed to complete a review article in this field which was our main focus. We accept other work exists with similar objectives. We hope with these further updates to the initial draft we have provided a more extensive review in this field.
Reviewer 3 Report
Comments and Suggestions for Authors
This is a well written review that describes the newly developed KRAS G12C inhibitors, sotorasib and adagrasib, on treatment of NSCLCs. The authors describe the benefits and adverse effects of these drugs on patients, and potential reasons for resistance development. The tables showing the relative efficacy and toxicity are helpful; here, inclusion of information on any ongoing clinical trials of sotorasib or adagrasib with immunotherapies (checkpoint inhibitors) and the efficacy as well as adverse events, if known, would improve this review significantly. Similarly, a schematic showing the potential resistance mechanism would be helpful.
Author Response
Many thanks for these comments. We appreciate this positive review. Based on this feedback we have included 2 further tables incorporating clinical trials in progress. This is an extensive field and adds considerable additional data to this review in both text and table form. We have elaborated further on resistance mechanisms based on data from the published clinical trials.
Reviewer 4 Report
Comments and Suggestions for Authors
This manuscript by O’Leary and colleagues assesses the current evidence for KRASG12C inhibitor medications in terms of efficacy and safety, as well as future directions for these therapies. Overall, this is a very comprehensive review. However, schematics and time scale should be added to simplify the reading. Moreover, the part describing resistance mechanisms should also discuss oncogenic fusion, gene level copy gain, activation of alternative pathway or histologic transformation.
Comments on the Quality of English Languagewell-written
Author Response
Many thanks for these comments. We appreciate this positive review. We have added an additional figure, tables and real-world data to the updated version. Our focus in this review article is predominantly on clinical data aimed towards the average medical oncologist in the clinic. We have touched on the science to provide context.
Round 2
Reviewer 1 Report
Comments and Suggestions for Authors
Thank you for upgrading the review article
Author Response
Many thanks for your helpful response. We have returned our manuscript with further updates for your review. Thank you for your ongoing feedback.
Reviewer 2 Report
Comments and Suggestions for Authors
The authors did not address my concerns.
Author Response
We apologize for our oversight. We have updated the article with further discussion on novel approaches looking at downregulation of KRAS signaling pathways in KRAS mutant malignancies, as well as a statement at the start of the paper outlining updates in our paper compared to prior works on this topic